# Antibiotic prescription practices in primary care in low- and middle-income countries: A systematic review and meta-analysis

Giorgia Sulis[1,2], Pierrick Adam[1,2], Vaidehi Nafade[1,2], Genevieve Gore[3], Benjamin Daniels[4], Amrita Daftary[2,5], Jishnu Das[4], Sumanth Gandra[6‡], Madhukar Pai[1,2,7‡]*

1 Department of Epidemiology, Biostatistics and Occupational Health, McGill University, Montreal, Quebec, Canada, 2 McGill International TB Centre, McGill University, Montreal, Quebec, Canada, 3 Schulich Library of Physical Sciences, Life Sciences and Engineering, McGill University, Montreal, Quebec, Canada, 4 McCourt School of Public Policy, Georgetown University, Washington, District of Columbia, United States of America, 5 School of Health Policy and Management, Faculty of Health, York University, Toronto, Ontario, Canada, 6 Division of Infectious Diseases, Department of Medicine, Washington University School of Medicine, St. Louis, Missouri, United States of America, 7 Manipal McGill Program for Infectious Diseases, Manipal Centre for Infectious Diseases, Manipal Academy of Higher Education, Manipal, Karnataka, India

‡ These authors are joint senior authors on this work.
* madhukar.pai@mcgill.ca

**Data Availability Statement:** All relevant data are within the manuscript and its Supporting Information files.

## Abstract

### Background

The widespread use of antibiotics plays a major role in the development and spread of antimicrobial resistance. However, important knowledge gaps still exist regarding the extent of their use in low- and middle-income countries (LMICs), particularly at the primary care level. We performed a systematic review and meta-analysis of studies conducted in primary care in LMICs to estimate the prevalence of antibiotic prescriptions as well as the proportion of such prescriptions that are inappropriate.

### Methods and findings

We searched PubMed, Embase, Global Health, and CENTRAL for articles published between 1 January 2010 and 4 April 2019 without language restrictions. We subsequently updated our search on PubMed only to capture publications up to 11 March 2020. Studies conducted in LMICs (defined as per the World Bank criteria) reporting data on medicine use in primary care were included. Three reviewers independently screened citations by title and abstract, whereas the full-text evaluation of all selected records was performed by 2 reviewers, who also conducted data extraction and quality assessment. A modified version of a tool developed by Hoy and colleagues was utilized to evaluate the risk of bias of each included study. Meta-analyses using random-effects models were performed to identify the proportion of patients receiving antibiotics. The WHO Access, Watch, and Reserve (AWaRe) framework was used to classify prescribed antibiotics. We identified 48 studies from 27 LMICs, mostly conducted in the public sector and in urban areas, and predominantly based on medical records abstraction and/or drug prescription audits. The pooled

**Funding:** The authors received no specific funding for this work.

**Competing interests:** I have read the journal's policy and the authors of this manuscript have the following competing interests: MP is a member of the Editorial Board of PLOS Medicine, and he co-edits the PLOS Tuberculosis Channel.

**Abbreviations:** AMR, antimicrobial resistance; AWaRe, Access, Watch, and Reserve; HIC, high-income country; LMICs, low- and middle-income countries; WHO, World Health Organization.

prevalence proportion of antibiotic prescribing was 52% (95% CI: 51%–53%), with a prediction interval of 44%–60%. Individual studies' estimates were consistent across settings. Only 9 studies assessed rationality, and the proportion of inappropriate prescription among patients with various conditions ranged from 8% to 100%. Among 16 studies in 15 countries that reported details on prescribed antibiotics, Access-group antibiotics accounted for more than 60% of the total in 12 countries. The interpretation of pooled estimates is limited by the considerable between-study heterogeneity. Also, most of the available studies suffer from methodological issues and report insufficient details to assess appropriateness of prescription.

## Conclusions

Antibiotics are highly prescribed in primary care across LMICs. Although a subset of studies reported a high proportion of inappropriate use, the true extent could not be assessed due to methodological limitations. Yet, our findings highlight the need for urgent action to improve prescription practices, starting from the integration of WHO treatment recommendations and the AWaRe classification into national guidelines.

## Trial registration

PROSPERO registration number: CRD42019123269.

---

## Author summary

### Why was this study done?

- Inappropriate use of antibiotics, both in terms of incorrect regimens and prescription without clinical indication, is a major driver of antibiotic resistance.

- Global drug sales data indicate a substantial increase in antibiotic use in low- and middle-income countries (LMICs) over the past 2 decades.

- An accurate quantification of antibiotic prescribing in primary care across LMICs is not available.

### What did the researchers do and find?

- We conducted a systematic review and meta-analysis to estimate the proportion of antibiotic prescribing across primary care settings in LMICs.

- Our study showed that, on average, approximately half of patients attending primary care facilities in LMICs received at least 1 antibiotic.

- Very few included studies made an attempt to assess the extent of inappropriate prescriptions and indicate potential misuse.

- Among studies that provided information on the types of antibiotics used, we found that, in 12/16 studies, 60% of prescriptions were for antibiotics with low potential for resistance selection as defined by the World Health Organization (WHO).

**What do these findings mean?**

- Our study highlights that antibiotics are highly prescribed in outpatient primary care settings.

- Better quality data are necessary to dig deeper into the patterns of inappropriate use according to local epidemiologic scenarios.

- Adapting WHO treatment recommendations and incorporating the WHO Access, Watch, and Reserve (AWaRe) classification of antibiotics into national guidelines will be a first key step to improve prescription practices.

## Introduction

Antimicrobial resistance (AMR) is a major health threat globally [1]. Growing morbidity and mortality rates due to resistant infections in humans are expected worldwide, along with a substantial economic impact in terms of productivity losses and healthcare expenditures [2,3].

Several factors are known to play a role in the development and spread of AMR, with inappropriate use of antibiotics being one of its most important drivers [4]. Gathering data about resistance as well as antibiotic use is 1 of the top 5 priorities of the Global Action Plan on Antimicrobial Resistance by the World Health Organization (WHO) [5].

A multinational survey conducted across 76 countries to determine the magnitude of antibiotic consumption and its trend over time revealed a dramatic increase between 2000 and 2015 (+65% globally), mostly driven by a sharp rise in low- and middle-income countries (LMICs) (+114%), where the levels of antibiotic consumption are high and rapidly approaching those observed in high-income countries (HICs) [6]. However, this analysis was based on drug sales data, thus providing limited information regarding providers' prescription habits.

The high level of antibiotic consumption in LMICs is because of multiple factors, including the high burden of infectious diseases, lack of regulations (or weak enforcement) to prevent over-the-counter sale of antibiotics, inadequate training of healthcare professionals, and the limited availability of essential diagnostics, which favors empirical use of antibiotics [1,7,8]. Besides misuse (i.e., prescription without clinical indication), another huge concern is the inappropriate use of antibiotics in terms of choice of a suitable molecule, dosage, and duration of treatment according to the site of infection and patient's characteristics.

Most studies investigating the magnitude and determinants of antibiotic use have focused on HICs, and those from LMICs have been carried out predominantly in hospital settings [9–12], leaving a number of unanswered questions about current practices at the primary healthcare level, where the bulk of antibiotic use takes place.

Of note, there is a paucity of information regarding the degree and pattern of antibiotic use in outpatient primary healthcare facilities, i.e., any service (other than pharmacies) providing care for people making an initial contact with a health professional. Having this information will be helpful to design and implement effective stewardship interventions and policies in LMICs.

We conducted a systematic review of the literature to assess the extent and patterns of antibiotic prescription and their determinants at the primary healthcare level in LMICs, as well as the proportion of such prescriptions deemed to be inappropriate.

## Methods

The protocol for this systematic review was registered in the International Prospective Register of Systematic Reviews (PROSPERO) (identifier: CRD42019123269) and followed the PRISMA guidelines [13]. The PRISMA checklist and PROSPERO protocol are provided as S1 PRISMA Checklist and S1 PROSPERO Protocol.

### Search strategy and selection criteria

We performed a systematic review of cross-sectional studies that were conducted in primary care in LMICs and reported the proportion of individuals receiving any antibiotic or the proportion of drug prescriptions that included an antibiotic. We also examined randomized and non-randomized trials as well as other observational studies to determine whether potentially relevant information (e.g., results from preliminary field assessments including cross-sectional drug prescription data) was provided. Conference proceedings and abstracts, commentaries, editorials, reviews, mathematical modeling studies, economic analyses, qualitative studies, and studies published in predatory journals as defined by Beall [14] were excluded. Studies conducted solely in an inpatient setting, those that focused on veterinary use of antibiotics, and those that only enrolled patients belonging to special cohorts (e.g., patients with cystic fibrosis or neutropenia or other underlying conditions that may justify an increased empirical use of antibiotics, or patients receiving antibiotics as part of prophylactic regimens) were also ineligible. No restrictions were applied with regards to the population characteristics in terms of age, sex, pregnancy status, or HIV status.

For the purpose of the study, we considered as "primary care" any care provided by any health professional (other than pharmacists) with whom patients have their initial contact, in the public or private sector, including primary care delivered in hospital settings wherever appropriate. In cases of uncertainty, we contacted the study authors for clarification. Antibiotics were defined as any agents included in the J01 group of the ATC (Anatomical Therapeutic Chemical) classification system [15]. Inappropriate prescriptions were recorded when such assessment was performed in the original studies. Countries were classified as low, lower-middle, upper-middle, or high income following the World Bank categorization based on gross national income per capita (GNI) of the study start year [16]. GNI thresholds for the definition of such categories, which have changed slightly over time, are provided in S1 Table. Given that there is no international standard definition of "urban" and "rural" areas, we classified the study settings in accordance with the authors' statements. If not explicitly stated by the investigators, we categorized as "urban" any site with a minimum population of 2,000 inhabitants, i.e., the most frequently used cutoff [17].

The search strategy was built in collaboration with a medical librarian (GG), using key terms for "antibiotic," "primary healthcare," "prescribing," and "LMICs" (both as a group and as individual countries, adopting a filter that was developed according to the World Bank categories). Medline (PubMed), Embase (Ovid), Global Health (Ovid), and CENTRAL (Cochrane Library) were systematically searched from 1 January 2010 until 4 April 2019. We also reran our search on 11 March 2020 using PubMed only; for feasibility reasons, the update could not be conducted through all data sources used in the initial search. Studies conducted before 1 January 2010 were excluded. The start date of our search was established after the conduction of an exploratory review of the literature showing that only a small number of studies were performed before 2010 in relevant settings, in the face of the exponentially higher number of total records identified through our search strategy, which would have posed substantial feasibility issues with very little benefit. Additionally, as patterns of antibiotic prescribing have changed substantially over time, including older studies would have been of limited value for

understanding the current situation. No language restrictions were applied. The full search strategies for each database are presented in S1 Text.

## Study screening and data extraction

Search results were imported into a citation manager (EndNote X9, Clarivate Analytics), and duplicates were removed. Three authors (GS, PA, and VN) independently screened citations by title and abstract against predefined eligibility criteria. The full-text review of all selected records was performed by 2 authors (GS and PA). An electronic data extraction form was piloted on 5 randomly selected papers and then used by 2 reviewers (GS and PA) to extract information from all eligible publications. At each stage of the screening and data extraction process, disagreements were resolved through discussion, and, if necessary, a third author (SG) was consulted to reach consensus. Study authors were contacted to request clarifications or additional data if needed. A detailed description of the screening and data extraction process is provided in S2 Text along with interrater agreement statistics.

## Assessment of study quality and publication bias

A modified version of a tool developed by Hoy and colleagues was utilized to evaluate the risk of bias of each included study (S2 Table) [18]. Our checklist included 8 methodological items (rated as low or high risk of bias), plus a summary item on the overall risk of study bias (rated as low, moderate, or high); no numeric scores were applied. All findings from this assessment were recorded in the data extraction form by the same independent reviewers. As a sensitivity analysis, we excluded studies with a high overall risk of bias.

No formal assessment of publication bias could be performed since traditional approaches such as funnel plots and tests for asymmetry are considered unsuitable for prevalence studies [19].

## Statistical analysis

Depending on the type of data available from individual studies, we calculated either the proportion of patients evaluated in a given health facility or by a certain provider who received antibiotics or the proportion of all drug prescriptions containing any antibiotics, along with their Clopper–Pearson (or exact) 95% confidence intervals (CIs) [20]. The 2019 WHO Access, Watch, and Reserve (AWaRe) framework was used to classify antibiotics according to their potential for selecting resistance [21]. Access-group antibiotics are first-line and narrow-spectrum agents such as penicillin, amoxicillin, and trimethoprim-sulfamethoxazole. Watch-group antibiotics are broad-spectrum agents with higher resistance selection such as second- and third-generation cephalosporins, and fluoroquinolones. Reserve-group antibiotics include last-resort antibiotics such as colistin. Fixed-dose combinations of antibiotics (e.g., ciprofloxacin/ornidazole) were classified as "discouraged" antibiotics, in line with WHO recommendations.

Random-effects meta-analyses were performed to estimate pooled proportions after Freeman–Tukey transformation to normalize the outcome [22]. To assess the between-study heterogeneity, we used the $I^2$ statistic and calculated prediction intervals (i.e., a type of confidence interval that provides the 95% range of true values to be expected in similar studies) [23,24]. Random-effects meta-regression with Knapp–Hartung adjustment (aimed to accommodate high degrees of heterogeneity) was employed to investigate the sources of heterogeneity. Categorical predictors for facility location (urban/rural), healthcare sector (public/private), age group (adults/children/all), type of patients (i.e., patients seeking care for any reason or individuals with a specific condition, e.g., diarrhea), and source of prescription information were

considered for building the model. If collinearity issues were observed, variables with the lowest number of missing values were prioritized and included in the model.

Subgroup analyses were conducted to investigate potential differences across levels of country income and types of patients involved (with a focus on studies where all patients attending 1 or more facilities were considered without placing restrictions based on their clinical presentation).

Sensitivity analyses were done by repeating analyses without studies that (i) were conducted in Iran as they were all based on administrative data from national registers; (ii) did not report details on the population and/or health facility location; (iii) were conducted in low-income countries; (iv) were based on the standardized patient methodology, in which antibiotics were deemed inappropriate by indication; (v) were deemed to be low quality (i.e., overall risk of study bias scored as "high").

All analyses were conducted in Stata (version 14; StataCorp) [25,26].

## Results

Our initial search yielded 9,604 unique citations, and an additional 590 were retrieved through our search update. A total of 48 studies (all cross-sectional) were finally included in the analyses (Fig 1) [27–74]. All included publications were in English language, except for 1 that was in Spanish. A summary of the main study characteristics is presented in Table 1, and the full dataset used for analyses is provided as S1 Data. Most studies were conducted in lower-middle- or upper-middle-income countries (22 and 19, respectively), while only 6 were in a low-income country. Additionally, 1 study was carried out in 3 countries (1 low income and 2 lower-middle income) [70]. Both public and private healthcare services were involved in 10 of the 48 (20.8%) included studies, whereas 26 (54.2%) studies were focused on the public sector, 4 (8.3%) were focused on the private sector, and 8 (16.7%) did not provide this information; none of the studies mentioned any involvement of informal practitioners. Facilities located in urban areas were more represented than those located in rural areas (17/48 studies [35.4%; 95% CI: 22.2%–50.5%] versus 10/48 studies [20.8%; 95% CI: 10.5%–35.0%]), with 13 (27.1%) studies involving both settings and 8 (16.7%) not reporting sufficient details. While 9 (18.8%) studies only included individuals presenting with 1 prespecified condition (i.e., acute respiratory illness, diarrhea, or fever), the other studies did not apply restrictions on the reason for seeking care and/or the final diagnosis (if any) and likely included patients with various conditions. None of the studies focused solely on dental care; although it is possible that patients seeking dental care were included in some studies, this group likely represented a negligible proportion of the total sample. Of note, no clinical information was reported in most studies.

Importantly, almost all the studies identified through our systematic review only assessed drug prescription and did not account for direct dispensing of unlabeled medicines, which is likely a common practice [75]. This may underestimate the true antibiotic prescribing proportion.

### Study quality

Fig 2 displays the summary of the risk of bias assessment, while the individual studies' quality assessment results are presented in S3 Table. The overall risk of study bias was scored as high for 21/48 studies (43.8%), moderate for 11 (22.9%), and low for 16 (33.3%). The proportion of studies assigned to the high risk group was higher among those conducted in low- and lower-middle-income countries (14/28; 50%) and lower among those performed in upper-middle-income countries (7/19; 36.8%). No major changes were observed in terms of overall study quality over time, although this could be due to the limited number of studies. In general, the

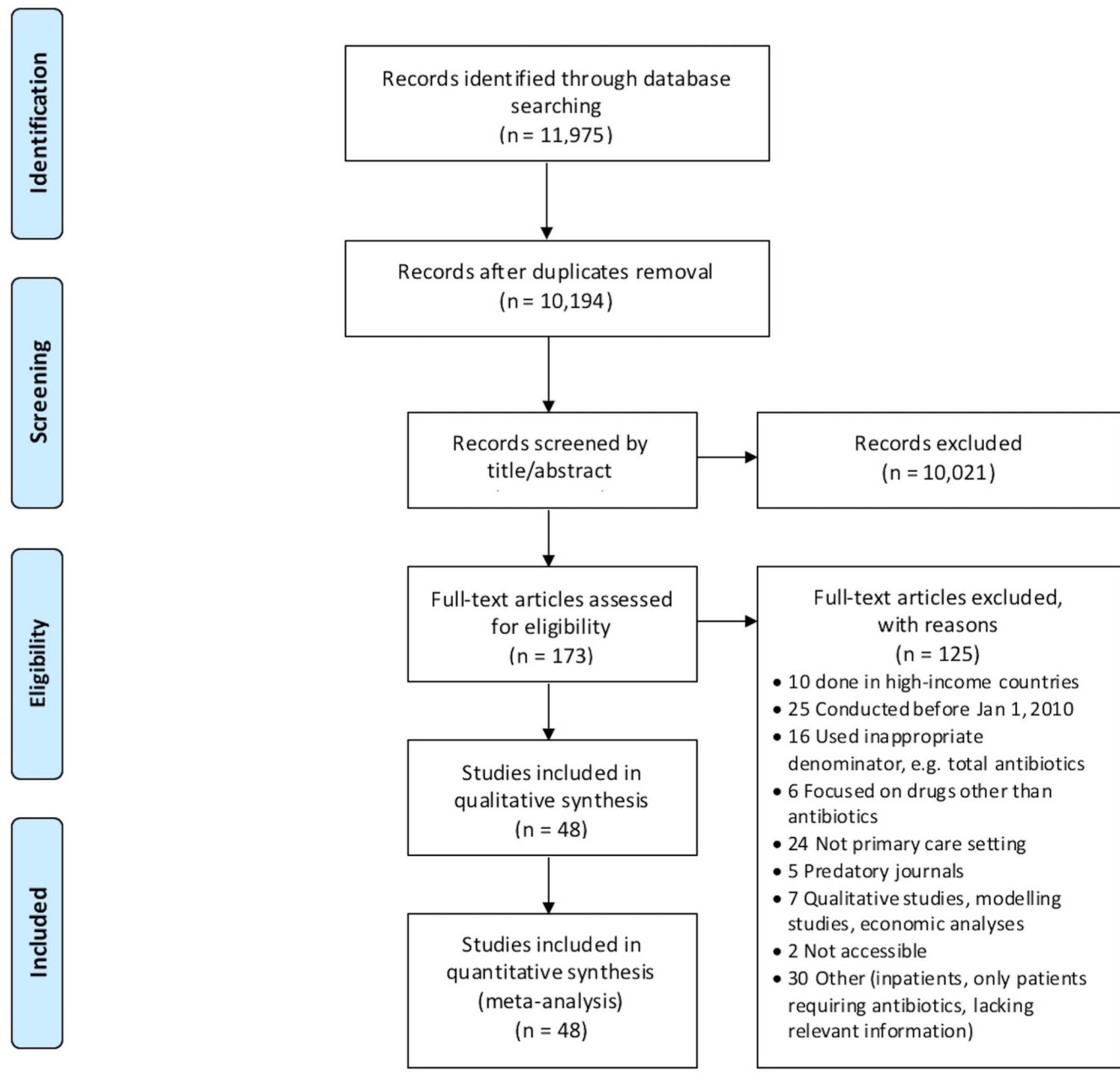

**Fig 1. PRISMA diagram.**

biggest issues were observed with regards to external validity: Some form of random sampling or a census was seldom performed, and the study population was rarely representative of the target, mostly due to the fact that prescriptions were often selected from one or a few facilities in circumscribed areas. The case definition was considered inadequate for studies that did not record clinical details about patients receiving prescriptions. The risk of bias concerning the data collection method was deemed to be low for studies that used medical records or similar sources to retrieve prescription information. This choice was made based on the fact that

**Table 1. Characteristics of studies identified through systematic review.**

| Income level | Study | Country | Health sector | Facility location | Number of facilities involved | Data source | Age group | Denominator* |
|---|---|---|---|---|---|---|---|---|
| Low | Baltzell 2019 [68] | Malawi | Private | Rural | NA | Medical records | NA | 9,924 (P) |
| | Mukonzo 2013 [27] | Uganda | Both | Both | 1 | Medical records, prescription audit | All | 173 (P) |
| | Nepal 2020 [73] | Nepal | Public | Urban | NA | Prescription audit | All | 950 (P) |
| | Savadogo 2014 [28] | Burkina Faso | Public | Urban | 2 | Medical records | Children | 376 (P) |
| | Worku 2018 [29] | Ethiopia | Public | Urban | 6 | Medical records, prescription audit | All | 898 (D) |
| | Yebyo 2016 [30] | Ethiopia | Public | Rural | 4 | Medical records | Adults | 414 (P) |
| Lower-middle | Abdulah 2019 [31] | Indonesia | Public | NA | 25 | Prescription audit | Adults | 10,118 (D) |
| | Adisa 2015 [32] | Nigeria | Public | Urban | 8 | Prescription audit | Adults | 400 (P) |
| | Ahiabu 2016 [33] | Ghana | Both | Both | 4 | Medical records | All | 1,600 (D) |
| | Akl 2014 [34] | Egypt | Public | Urban | 10 | Medical records | NA | 1,000 (D) |
| | Atif 2016 [35] | Pakistan | NA | Urban | 10 | Prescription audit | NA | 1,000 (D) |
| | Beri 2013 [36] | India | Private | Urban | 20§ | Provider interview | All | 400 (P) |
| | Chem 2018 [37] | Cameroon | Both | Both | 26 | Medical records | All | 30,096 (D) |
| | El Mahalli 2011 [38] | Egypt | Public | Urban | 2 | Medical records | Children | 300 (P) |
| | Graham 2016 [39] | Zambia | NA | NA | 90§ | Provider interview | Children | 537 (P) |
| | Jose 2016 [40] | India | Public | Rural | 1 | Prescription audit | Children | 552 (D) |
| | Kasabi 2015 [41] | India | Public | NA | 20 | Medical records | NA | 600 (P) |
| | Mekuria 2019 [72] | Kenya | Private | Urban | 4 | Prescription audit | All | 17,382 (P) |
| | Ndhlovu 2015 [42] | Zambia | Both | Both | 148 | Patient interview, medical records | All | 872 (P) |
| | Omole 2018 [43] | Nigeria | Both | Rural | NA | Prescription audit | NA | 4,255 (D) |
| | Oyeyemi 2013 [44] | Nigeria | Public | Urban | 4 | Medical records | All | 600 (D) |
| | Raza 2014 [45] | Pakistan | Both | Urban | NA | Prescription audit | NA | 1,097 (D) |
| | Sarwar 2018 [46] | Pakistan | Public | Both | 32 | Prescription audit | NA | 6,400 (D) |
| | Saurabh 2011 [47] | India | NA | Rural | 4 | Prescription audit | NA | 600 (D) |
| | Saweri 2017 [48] | PNG | Public | Both | 7 | Ad hoc form | All | 6,008 (P) |
| | Sudarsan 2016 [49] | India | Public | Urban | 1 | Prescription audit | NA | 360 (D) |
| | Yousif 2016 [50] | Sudan | Both | NA | 220§ | Prescription audit | NA | 19,690 (D) |
| | Yuniar 2017 [51] | Indonesia | Both | NA | 56 | Prescription audit | NA | 1,657 (D) |
| Upper-middle | Ahmadi 2017 [52] | Iran | Public | Rural | 103 | Prescription audit | NA | 352,399 (D) |
| | Alabid 2014 [53] | Malaysia | Private | Urban | 70 | Patient interview | Adults | 140 (P) |
| | Bielsa-Fernandez 2016 [54] | Mexico | NA | Urban | 109§ | Provider interview | All | 1,840 (P) |
| | Gasson 2018 [55] | South Africa | Public | Urban | 8 | Medical records | All | 654 (P) |
| | Greer 2018 [56] | Thailand | Public | Both | 32 | Medical records | All | 83,661 (P) |
| | Lima 2017 [57] | Brazil | NA | NA | 20 | Prescription audit | NA | 399 (D) |
| | Liu 2019 [71] | China | Public | Both | 65 | Prescription audit | All | 428,475 (D) |
| | Mashalla 2017 [58] | Botswana | Public | Urban | 19 | Prescription audit | All | 550 (D) |
| | Ab Rahman 2016 [59] | Malaysia | Both | Both | 545 | Medical records | All | 27,587 (P) |
| | Sadeghian 2013 [60] | Iran | NA | NA | NA | Prescription audit | NA | 4,940,767 (D) |
| | Safaeian 2015 [61] | Iran | NA | Both | 3,772§ | Prescription audit | NA | 7,439,709 (D) |
| | Sánchez Choez 2018 [62] | Ecuador | Public | Both | 1 | Prescription audit | All | 1,393 (P) |
| | Sun 2015 [63] | China | Public | Both | 24 | Prescription audit | All | 1,468 (D) |
| | Wang 2014 [64] | China | Public | Both | 48 | Medical records | All | 7,311 (D) |
| | Xue 2019 [65] | China | Public | Rural | NA | SP exit interview | All | 526 (P) |
| | Yin 2015 [66] | China | Both | Urban | 2,501 | Prescription audit | NA | 42,200 (D) |
| | Yin 2019 [74] | China | Public | Rural | 8 | Prescription audit | All | 14,526 (D) |
| | Zhan 2019 [69] | China | Public | Rural | 17 | Prescription audit | All | 1,720 (D) |
| | Zhang 2017 [67] | China | Public | Rural | 20 | Prescription audit | Children | 9,340 (D) |

(*Continued*)

**Table 1.** (Continued)

| Income level | Study | Country | Health sector | Facility location | Number of facilities involved | Data source | Age group | Denominator* |
|---|---|---|---|---|---|---|---|---|
| Multiple | Kjærgaard 2019 [70] | Kyrgyzstan, Uganda, Vietnam | NA | NA | NA | Medical records, provider interview | Children | 699 (P) |

*Denominator used to calculate the outcome (i.e., total number of patients evaluated [P] or total number of drug prescriptions [D]).

§Number of healthcare providers involved.

NA, not available; PNG, Papua New Guinea; SP, standardized patient.

medical records and drug prescription audits constitute good sources to estimate the proportion of antibiotic prescribing, although they are generally poorly suited for an accurate evaluation of appropriateness of prescription. On the other hand, studies using patient or provider questionnaires were considered at high risk of bias given the potential for recall bias and Hawthorne effect [76,77].

## Prevalence of antibiotic prescription

Among the 21 studies that reported the total number of patients attending a certain facility at the time of data collection [27,28,30,32,36,38,39,41,42,48,53–56,59,62,65,68,70,72,73], the average proportion of individuals receiving an antibiotic prescription ranged widely, from 19.6% (95% CI: 14.0%–26.4%) to 90.8% (95% CI: 89.3%–92.0%) [27,54]. Among the 27 studies in which the denominator was the total number of drug prescriptions [29,31,33–35,37,40,43–47,49–52,57,58, 60,61,63,64,66,67,69,71,74], the proportion of prescriptions containing

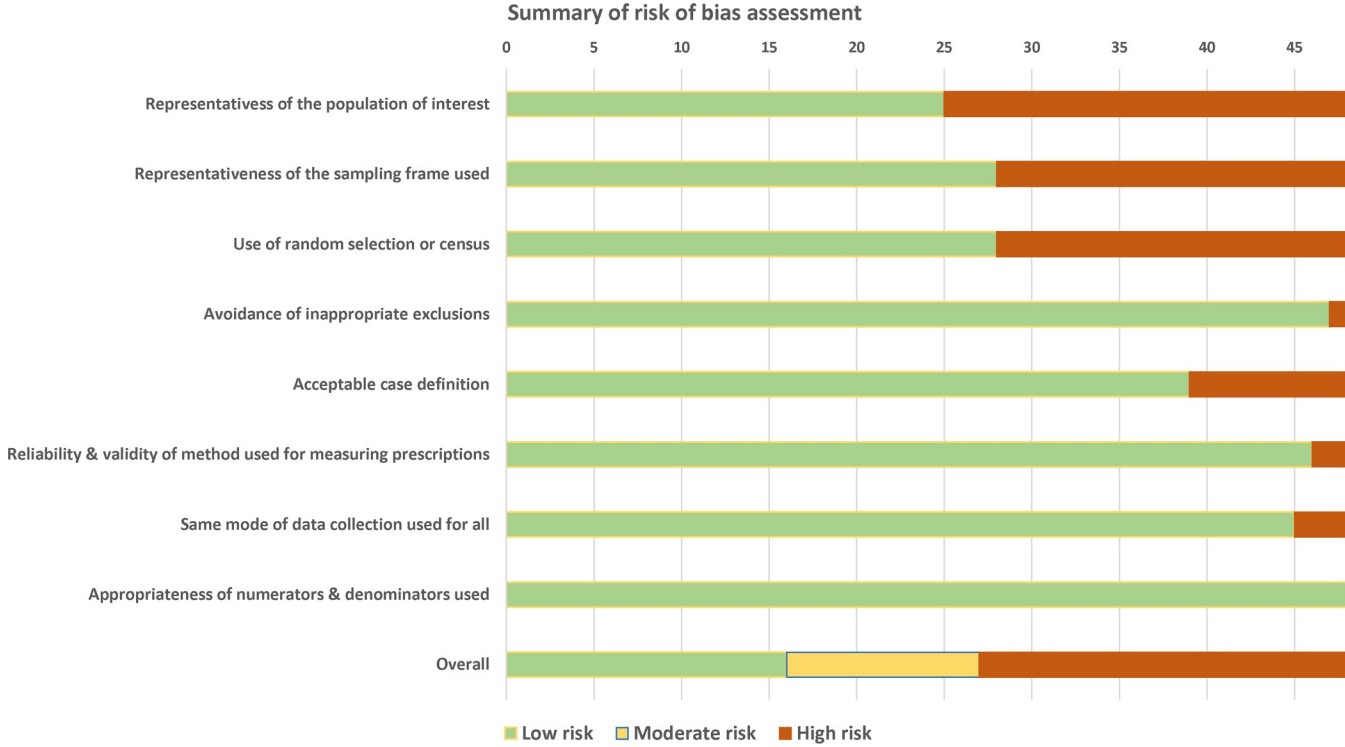

**Fig 2. Summary of study risk of bias assessment.**

antibiotics varied between 17.8% (95% CI: 14.2%–21.9%) and 79.2% (95% CI: 74.4%–82.7%) [46,57]. We could not identify any specific pattern in the distribution of antibiotic prescription rates across levels of country income, partly due to small sample sizes. As very few studies were conducted solely in the private health sector, no comparisons could be made against public facilities. Similar considerations apply to the health service location (i.e., urban versus rural areas). Furthermore, we did not observe any specific variation over time in the proportion of patients receiving antibiotics, either overall or after stratifying by country income level.

Since almost all patient–provider encounters included in studies using patients as the denominator resulted in a treatment prescription, prevalence estimates can be considered comparable to those derived from the 27 studies using drug prescriptions as the denominator. The pooled proportion of patients who received antibiotics resulting from a meta-analysis of all studies was 52% (95% CI: 51%–53%), and both stratum-specific pooled proportions for studies using one or the other type of denominator were reasonably close to the overall estimate (Fig 3). As expected, very high levels of between-study heterogeneity were observed ($I^2$ values were above 98% overall, in subgroup analyses, and in sensitivity analyses), thus limiting the reliability of our pooled estimates. However, the 95% prediction interval calculated in the primary analysis was quite narrow, ranging from 44% to 60%, indicating that a new potential observation in a similar setting would likely yield a proportion of patients receiving antibiotics close to 50%. The prediction interval is wider than the conventional confidence interval owing to the fact that it accounts for uncertainty about both the population mean and the distribution of values.

Subgroup analyses (e.g., after stratification by country income level, type of denominator, or type of patients examined) and sensitivity analyses yielded similar point estimates, but confidence and prediction intervals became much wider (S1–S4 Figs). Unsurprisingly, given the results of subgroup meta-analyses, none of the coefficients of our meta-regression models was statistically significant, and the overall model could only explain a negligible proportion of the observed heterogeneity (S4 Table).

## Inappropriate antibiotic prescription

As previously mentioned, we recorded the proportion of inappropriate prescriptions when available in individual studies. In most cases, the authors made their judgment based on national and/or international guidelines for treatment of key conditions. Among the 9 studies that assessed the rationality of antibiotic prescriptions [36,39,46,53,55,62,64,65,67], the proportion judged inappropriate ranged widely, reflecting the significant differences in study designs as well as in the sets of criteria that were adopted to determine the outcome (Table 2). The lowest level of inappropriate prescription (7.9%; 95% CI: 4.6%–12.5%) was reported in a study conducted in Zambia that included 537 children aged <5 years presenting with an acute respiratory syndrome, of whom 37.6% (95% CI: 33.5%–41.9%) were given antibiotics [39]. All antibiotic prescriptions were classified as inappropriate in 3 studies: 2 of them employed standardized patients portraying conditions that did not require antibiotics such as common cold, watery diarrhea, presumptive tuberculosis, and chest pain indicative of angina, with an overall antibiotic prescription prevalence of about 41%–42% [53,65]; the other study was performed in China and included 9,340 drug prescriptions issued for children with acute respiratory tract infection of likely viral etiology, 36.6% (95% CI: 35.7%–37.6%) of whom received an antibiotic [67]. The proportion of inappropriate antibiotic prescriptions exceeded 50% in the remaining 5 studies.

Information regarding individual antibiotics was available from 16 studies in 15 countries. Of note, 11 of these studies included patients seeking care for any reason, while the remaining

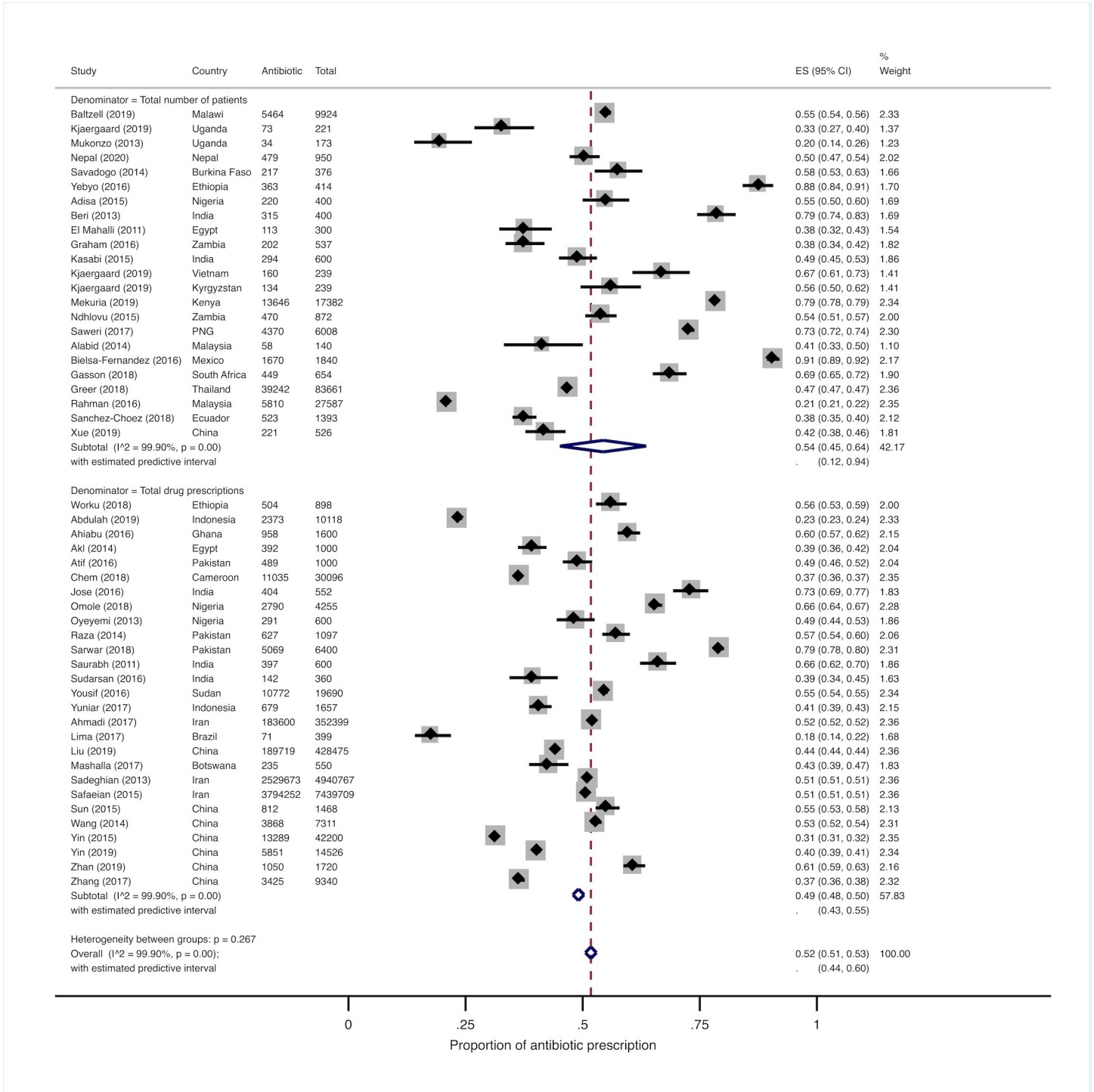

**Fig 3. Forest plot of antibiotic prescription prevalence across all studies stratified by type of denominator used (i.e., either total number of patients or total number of drug prescriptions).** CI, confidence interval; ES, effect size; PNG, Papua New Guinea.

5 studies focused on a specific condition (i.e., respiratory tract infection [4 studies] or diarrhea [1 study]) (Table 3). Access-group antibiotics accounted for the majority of prescriptions (more than 60%) in 13 studies from 12 countries, whereas Watch-group antibiotics accounted

**Table 2. Main findings of studies that assessed inappropriate antibiotic prescription.**

| Study | Country | Country income | Healthcare sector | Sample size | Type of patients | Antibiotic prescriptions n (%; 95% CI) | Inappropriate antibiotic prescriptions n (%; 95% CI) |
|---|---|---|---|---|---|---|---|
| Beri (2013) [36] | India | Lower-middle | Private | 400 | Patients of all ages with any clinical presentation | 315 (78.8; 74.4–82.7) | 179 (56.8; 51.2–62.4) |
| Graham (2016) [39] | Zambia | Lower-middle | Not reported | 537 | Children under age 5 years with acute respiratory illness | 202 (37.6; 33.5–41.9) | 16 (7.9; 4.6–12.5) |
| Sarwar (2018) [46] | Pakistan | Lower-middle | Public | 6,400 | Patients with any clinical presentation | 5,069 (79.2; 78.2–80.2) | 4,238 (83.6; 82.6–84.6) |
| Gasson (2018) [55] | South Africa | Upper-middle | Public | 654 | Patients with any clinical presentation | 449 (68.7; 64.9–72.2) | 305 (67.9; 63.4–72.2) |
| Sánchez Choez (2018) [62] | Ecuador | Upper-middle | Public | 1,393 | Patients of all ages with upper respiratory tract infection | 523 (37.5; 35.0–40.1) | 472 (90.2; 87.4–92.7) |
| Wang (2014) [64] | China | Upper-middle | Public | 7,311 | Patients of all ages with any clinical presentation | 3,868 (52.9; 51.8–54.1) | 2,344 (60.6; 59.0–62.1) |
| Alabid (2014) [53] | Malaysia | Upper-middle | Private | 140 | Adult SPs with common cold | 58 (41.4; 33.2–50.1) | 58 (100) |
| Xue (2019) [65] | China | Upper-middle | Public | 526 | Adult and child SPs with 1 of the following: diarrhea (viral gastroenteritis), chest pain (suspicious for angina), fever and cough (presumptive TB) | 221 (42.0; 37.8–46.4) | 221 (100) |
| Zhang (2017) [67] | China | Upper-middle | Public | 9,340 | Children with upper respiratory tract infection | 3,425 (36.7; 35.7–37.7) | 3,425 (100) |

CI, confidence interval; SP, standardized patient; TB, tuberculosis.

for high proportions of prescriptions among studies from Mexico (90.3%; 95% CI: 88.8%–91.7%), China (78.4%; 95% CI: 75.7%–81.0%), and Pakistan (47.8%; 95% CI: 46.5%–49.1%) (Table 3) [46,54,63].

## Discussion

To our knowledge, this is the first comprehensive analysis of antibiotic prescriptions in primary care in LMICs. We found that the proportion of patients seeking care for any reason who were prescribed antibiotics in this context often exceeded 50%. Although the interpretation of our pooled estimates is limited by the considerable between-study heterogeneity, values were quite consistent across settings. Available studies from LMICs often suffer from several methodological issues and report scanty details concerning patients' clinical features that would help accurately judge the appropriateness of prescription. The number of health facilities involved in individual studies is often very small, particularly in low-income countries (a total of 13 facilities across 4 studies that reported this information), indicating major discrepancies in the quality of information among geographic areas. Although all included studies examined prescription data in primary care facilities, we recognize that primary care entails a wide range of facility types, each with its own peculiarities and challenges. This variegated scenario prevented us from conducting specific subgroup analyses that could inform targeted antibiotic stewardship strategies. Two studies, both conducted in an Iranian province, had a very large sample size because prescription details were captured through an electronic data collection system that is available nationwide. However, clinical information on patients receiving each prescription is much more challenging to obtain from this system, thus hindering a thorough assessment of inappropriate drug use.

**Table 3. AWaRe classification of antibiotic prescriptions in a subset of studies included in analysis.**

| Study, total number (n) of antibiotics prescribed or dispensed | Country | Patients' clinical presentation | Access-group antibiotics (%) | Watch-group antibiotics (%) | Reserve-group antibiotics (%) | Discouraged antibiotics (%) |
|---|---|---|---|---|---|---|
| Abdulah (2019) [31], n = 2,389 | Indonesia | Any | 1,667 (69.8) | 287 (12.0) | NA | NA |
| Sarwar (2018) [46], n = 5,853 | Pakistan | Any | 3,055 (52.2) | 2,798 (47.8) | 0 | 0 |
| Sánchez Choez (2018) [62], n = 553 | Ecuador | Acute respiratory syndrome | 463 (83.7) | 90 (16.3) | 0 | 0 |
| Worku (2018) [29], n = 553 | Ethiopia | Any | 431 (77.9) | 122 (22.1) | 0 | 0 |
| Gasson (2018) [55], n = 519 | South Africa | Any | 361 (69.6) | 158 (30.4) | 0 | 0 |
| Chem (2018) [37], n = 12,350 | Cameroon | Any | 11,109 (90.0) | 1,241 (10.0) | 0 | 0 |
| Mashalla (2017) [58], n = 289 | Botswana | Any | 240 (83.0) | 49 (17.0) | 0 | 0 |
| Ab Rahman (2016) [59], n = 6,009 | Malaysia | Any | 3,879 (64.6) | 2,073 (34.5) | NA | NA |
| Adisa (2015) [32], n = 303 | Nigeria | Any | 224 (73.9) | 61 (20.1) | 0 | 18 (5.9) |
| Yebyo (2016) [30], n = 373 | Ethiopia | Acute respiratory syndrome | 312 (83.6) | 61 (16.4) | 0 | 0 |
| Ndhlovu (2015) [42], n = 561 | Zambia | Any | 490 (87.3) | 42 (7.5) | 0 | 0 |
| Sun (2015) [63], n = 978 | China | Acute respiratory syndrome | 174 (17.8) | 767 (78.4) | NA | NA |
| Bielsa-Fernandez (2016) [54], n = 1,718 | Mexico | Diarrhea | 166 (9.7) | 1,551 (90.3) | 1 (0.06) | 0 |
| Mukonzo (2013) [27], n = 9,683 | Uganda | Any | 7,735 (79.9) | 1,908 (19.7) | NA | NA |
| Nepal (2020) [73], n = 479 | Nepal | Any | 299 (62.4) | 165 (34.4) | NA | NA |
| Mekuria (2019) [72], n = 13,646 | Kenya | Acute respiratory syndrome | 8,461 (62.0) | 4,880 (35.7) | NA | 278 (2.0) |

Denominator for percentage calculations is the total number of antibiotics dispensed/prescribed. Access-group antibiotics are first-line and narrow-spectrum agents such as penicillin, amoxicillin, and trimethoprim-sulfamethoxazole. Watch-group antibiotics are broad-spectrum agents with higher resistance selection such as second- and third-generation cephalosporins, and fluoroquinolones. Reserve-group antibiotics include last-resort antibiotics such as colistin. Discouraged antibiotics are fixed-dose combinations such as ciprofloxacin/ornidazole.

NA, not available.

WHO recommends that the proportion of patients receiving antibiotics in an outpatient setting should be less than 30% [78]. However, this threshold was established somewhat arbitrarily more than 2 decades ago, due to a lack of evidence on prescription practices and actual needs according to patients' clinical features. If accurate and nationally representative prescribing data were available for individual countries, these could be used as a benchmark to define condition-specific ideal prescribing proportions that account for context-related variables.

High infectious disease burden in LMICs could potentially explain the high prevalence of antibiotic use; however, our results raise concerns about potential misuse of antibiotics based on a subset of studies that assessed the rationality of antibiotic prescriptions. For example, high levels of antibiotic prescriptions (41%–42%) were reported in 2 standardized patient studies in Malaysia and China, where nobody should have received antibiotics, by design [53,65]. In a study conducted in Mexico, 69% of patients had had watery diarrhea for less than 48 hours, but almost everybody received antibiotics instead of rehydration alone [54]. Similarly, in a nationwide health facility survey in Zambia, 72.2% of patients met the criteria for suspected malaria, for which antibiotics are not appropriate treatment, but nonetheless more than half were given antibiotics [42]. Studies focused on individuals with upper respiratory symptoms such as common cold or pharyngitis reported unacceptably high antibiotic prescribing proportions, ranging from 36.7% to 55.3% [39,62,63,67].

To promote the optimal use of antibiotics and assist antibiotic stewardship efforts, WHO introduced the AWaRe classification in 2017 [21]. The classification underlines that, where appropriate, narrow-spectrum antibiotics included in the Access group should be preferred over broad-spectrum antibiotics from Watch and Reserve groups in order to limit the selection and spread of antibiotic resistance. Accordingly, WHO recommends that Access-group antibiotics should constitute at least 60% of overall antibiotic use [21]. Only 16 of the 48 studies identified through our systematic review reported detailed information on individual antibiotic drugs, and all but 3 had at least 60% of antibiotics being from the Access group [21]. Three studies with a high proportion of Watch-group antibiotics were from Mexico, China, and Pakistan; however, we cannot generalize these estimates to overall antibiotic consumption in these countries based on only 1 study in each country. Interestingly, a recent study that analyzed pediatric antibiotic sales data using AWaRe categories in 70 countries showed a high proportion of Watch-group antibiotics in China, Pakistan, and Mexico [79].

A recently published umbrella review on antibiotic use for adults in primary care (though focused on dental care) identified several factors that appear to affect prescribing behaviors in HICs, such as socio-cultural context, financial incentives, personal beliefs, patients' attitudes, and AMR awareness [80]. Similar considerations likely apply to prescription practices in LMICs, although a deeper understanding of underlying determinants remains challenging. Among the biggest issues is the poor documentation of clinical reasons leading to antibiotic prescription, as observed in other settings [81]. Reaching a definitive diagnosis is often a huge challenge in resource-constrained areas, where point-of-care diagnostic tests for the most common conditions observed in primary care are frequently lacking [82].

Along with potential antibiotic misuse, therapeutic schemes may be inappropriate because of inadequate choice of antibiotic or incorrect dose or duration. However, a thorough assessment of prescription practices that includes such considerations is made particularly difficult by the variability in national treatment guidelines regarding antibiotic regimens [83]. In an attempt to foster the harmonization of such guidelines and minimize differences across countries, WHO recently released antibiotic treatment guidelines for 26 common infectious syndromes encountered in primary care and inpatient settings [84]. These guidelines currently indicate when and what antibiotics should be prescribed, and further work on harmonizing dose, duration, and formulation is ongoing [21].

In summary, the pooled estimate of antibiotic prescription in primary care settings across LMICs was 52%, but there was significant between-study heterogeneity. Further, the true extent of misuse was hard to discern, given the lack of data on appropriateness and the low quality of studies included. Future studies should use methodologies such as standardized patients, where the diagnosis is fixed by design, or include thorough laboratory testing to match diagnoses with antibiotic use. Accurate prescription audit tools are difficult to implement in most LMICs owing to the limited availability of electronic records. Also, the paucity of clinical details that can be captured through medical records (paper-based or not) makes it even harder to determine the appropriateness of prescription [85].

There is a need for better quality data to accurately measure the magnitude of antibiotic prescribing and dispensing by healthcare professionals at the primary care level accounting for local epidemiologic patterns. Global burden of disease data [86] combined with nationally representative AMR surveillance data [87] could be utilized to estimate the amount and type of antibiotics needed in a country, which could then be compared with existing national antibiotic consumption databases [6]. Meanwhile, LMICs should adapt the WHO infection treatment guidelines and incorporate the AWaRe categorization into their national antibiotic treatment guidelines to improve antibiotic prescribing. This will help countries to prioritize

surveillance and stewardship efforts aimed at curbing the spread of AMR and preserving the efficacy of currently available antibiotics.

## Supporting information

**S1 Data. Dataset used for analyses.**
(XLSX)

**S1 Fig. Forest plot of proportion of patients receiving antibiotics, restricted to studies including patients seeking care for any reason.**
(TIF)

**S2 Fig. Forest plot of proportion of patients receiving antibiotics stratified by country income level (LIC = low-income country; LMIC = lower-middle-income country; UMIC = upper-middle-income country).**
(TIF)

**S3 Fig. Forest plot of proportion of patients receiving antibiotics, including all studies except those conducted in Iran.**
(TIF)

**S4 Fig. Forest plot of proportion of patients receiving antibiotics, excluding studies whose overall risk of bias was scored as "high."**
(TIF)

**S1 PRISMA Checklist.**
(DOC)

**S1 PROSPERO Protocol.**
(PDF)

**S1 Table. World Bank criteria for the definition of countries' income level 2010–2018.**
(DOCX)

**S2 Table. Risk of bias assessment tool (adapted from Hoy et al. [18]).**
(DOCX)

**S3 Table. Risk of bias assessment of all studies included in final synthesis.**
(DOCX)

**S4 Table. Results of meta-regression analysis.**
(DOCX)

**S1 Text. Search strategies employed.**
(DOCX)

**S2 Text. Selection process and data extraction.**
(DOCX)

## Author Contributions

**Conceptualization:** Giorgia Sulis, Jishnu Das, Sumanth Gandra, Madhukar Pai.

**Data curation:** Giorgia Sulis, Pierrick Adam, Vaidehi Nafade, Genevieve Gore, Sumanth Gandra.

**Formal analysis:** Giorgia Sulis, Sumanth Gandra.

**Methodology:** Giorgia Sulis, Pierrick Adam, Vaidehi Nafade, Genevieve Gore, Benjamin Daniels, Amrita Daftary, Jishnu Das, Madhukar Pai.

**Resources:** Madhukar Pai.

**Software:** Benjamin Daniels.

**Supervision:** Jishnu Das, Sumanth Gandra, Madhukar Pai.

**Validation:** Giorgia Sulis.

**Visualization:** Giorgia Sulis.

**Writing – original draft:** Giorgia Sulis, Sumanth Gandra.

**Writing – review & editing:** Giorgia Sulis, Pierrick Adam, Vaidehi Nafade, Genevieve Gore, Benjamin Daniels, Amrita Daftary, Jishnu Das, Sumanth Gandra, Madhukar Pai.

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
