## [Editor Report · Decision Letter 0]

29 Jan 2020

Dear Dr Pai, 

Thank you for submitting your manuscript entitled "Antibiotic prescription practices in primary care in low- and middle-income countries: a systematic review and meta-analysis" for consideration by PLOS Medicine.

Your manuscript has now been evaluated by the PLOS Medicine editorial staff [as well as by an academic editor with relevant expertise] and I am writing to let you know that we would like to send your submission out for external peer review.

Kind regards,

Adya Misra, PhD,

Senior Editor

PLOS Medicine

---

## [Decision Letter · Decision Letter 1]

6 Mar 2020

Dear Dr. Pai,

Thank you very much for submitting your manuscript "Antibiotic prescription practices in primary care in low- and middle-income countries: a systematic review and meta-analysis" (PMEDICINE-D-20-00248R1) for consideration at PLOS Medicine. 

[LINK]

In light of these reviews, I am afraid that we will not be able to accept the manuscript for publication in the journal in its current form, but we would like to consider a revised version that addresses the reviewers' and editors' comments. Obviously we cannot make any decision about publication until we have seen the revised manuscript and your response, and we plan to seek re-review by one or more of the reviewers. 

We expect to receive your revised manuscript by Mar 27 2020 11:59PM. Please email us (plosmedicine@plos.org) if you have any questions or concerns.

We look forward to receiving your revised manuscript. 

Sincerely,

Adya Misra, PhD

Senior Editor 

PLOS Medicine

plosmedicine.org

Abstract- the last sentence of the methods and findings section should include a limitation of your study design 

Abstract- perhaps structure abstract according to PRISMA guidelines

Literature search- Please update this to current date, as we are coming up to a year since the last search

Reviewers have noted the definition of primary care must be defined for clarity, please include this in the introduction and methods section

PRISMA checklist- We ask that you include a sentence in your methods section that the study followed PRISMA guidelines and that a completed checklist has been provided as SI file xx. In addition, please use paragraphs and sections instead of page numbers as they are likely to change

Methods- if all study types were included in the search this sentence is confusing “We performed a systematic review of cross-sectional studies”. Please correct and clarify as needed

Please provide a rationale for excluding studies from “predatory” journals, especially as lists of predatory publishers have been deemed controversial and biased

Methods- please provide a brief summary of how antibiotic prescriptions were deemed to be inappropriate, providing references to tools/checklists as needed 

The conclusions need to be toned down, since the reliability of the pooled estimates is low owing to high heterogeneity between included studies

Comments from the reviewers:

Reviewer #1: This study reviews and meta-analyses antibiotic prescription practices in primary care in low- and middle income countries. While it is an interesting topic, I do think the manuscript should be improved before it would be acceptable for publication. 

1. Abstract. It is unclear why the authors focus on the prevalence of antibiotic prescriptions. Prevalence gives the proportion of a population who have a specific characteristic in a given time period, e.g. the proportion of the population that is using antibiotics at a given day. It would be more interesting to look at the incidence of antibiotic prescriptions or e.g. of those presenting with symptom x, what proportion of patients gets an antibiotic. 

2. I don't understand why the authors try to model a pooled estimate across all years and countries. Surely there are strong trends among prescribing trends in LMIC (which probably also differ per region), as also highlighted by the authors in the introduction. Thus the pooled estimate is impossible to interpret and highly dependent on the number of studies and samples sizes in different periods. 

3. Abstract. The proportion of prescriptions ranged from 8 to 100%. Surely this 100% is not correct, a number that is obtained using questionable methods. Given this, please also add a comment on the quality of the included studies. 

4. Introduction. 'An estimated 10 million deaths per year are expected to….' No, they are not expected to die, there are many problems with the report that came up with this number. See, for example https://www.ncbi.nlm.nih.gov/pmc/articles/PMC5127510/

5. Introduction. 'The high level of antibiotic consumption in LMICS…' Is it hight, or is it just coming closer to HIC?

6. 'Most studies investigating the magnitude and determinants of antibiotic use have been focused on high-income countries (HIC) and particularly on hospital settings.' While the first claim is true, there are many studies estimating the magnitude and determinants of (inappropriate) antibiotic use in primary care/outpatient setting, e.g.

https://www.bmj.com/content/364/bmj.l440.full

https://academic.oup.com/jac/article/73/suppl_2/ii36/4841818

https://www.sciencedirect.com/science/article/pii/S2589537018300531

https://academic.oup.com/jac/article/73/suppl_2/ii27/4841819

https://www.bmj.com/content/362/bmj.k3155

https://www.tandfonline.com/doi/full/10.1080/02813432.2017.1288680

In fact, in the UK we probably have a better understanding of prescribing practice than in the hospital setting.

Therefore, the sentence of paucity of information of antibiotic use in outpatient primary healthcare facilities, only applies potentially to LIMC. 

7. Related to above, if there truly is a paucity of information, there is no point in performing a systematic review. In that case it may be better to invest time in original research and collecting new data. 

8. Please clarify why papers published in predatory journals were excluded without looking at the actual quality of the paper. Wouldn't it be possible that a useful high-quality paper would be published in one of these journals, especially if open access fees may be lower than for other journals for authors from LMIC? 

9. '…as urban any site with a minimum population of 2,000 inhabitants..' This is per what? Per square mile/km? If it is 2000 persons in a very large geographical area, that area shouldn't be classified as urban. 

10. I have my doubts about the search term 'primary healthcare'. I think this term wouldn't pick up many papers that were performed in primary care/outpatient/community setting. It is also unclear why the authors haven't used MeSH terms as well. 

11. When discussing the reason for limiting to studies preformed >=2010, the authors should also add that in contrast to the effect of interventions which can usually be estimated using studies from different years, meta-analysing something that is almost guaranteed to chance over time, especially given the knowledge we have about trends in antibiotic prescribing, it doesn't make even sense to include relatively old studies in a pooled estimate. 

12. Please specify missing % for the different variables

13. Summarising inappropriateness proportions across very different conditions, that likely have different 'ideal' prescribing proportions, e.g. see https://academic.oup.com/jac/article/73/suppl_2/19/4841820

https://academic.oup.com/jac/article/73/suppl_2/ii11/4841821, isn't very informative. Separate estimates should be provided for different conditions. 

14. The study seems to be comparing apples and oranges with 8 studies including only subjects presenting with one pre-specified condition (with likely different ideal prescribing proportions see comment above) and other studies that did not apply any restrictions on reason for seeking care. These studies are not comparable and shouldn't be combined. 

15. Why report confidence intervals for study bias?? I don't think confidence intervals make sense to report here, it would be sufficient to just report the percentages without CI. 

16. I don't understand why the author mention a formal comparison of proportion of high-risk studies across different types of countries? Why would this be of interest?

17. '…. And the study population was rarely representative of the target….' The target of the current study, or the target of the original study?

18. Please add a reference for the Hawthorne effect.

19. 'the proportion of prescriptions containing antibiotics varied between 17.8% .. and 79%..' This is so dependent on other drugs and indications, hardly informative at all…

20. 'we could not identify any specific pattern in the distribution of antibiotic rates across levels of country income'. Could be added that this may be due to small sample sizes. 

21. I2 values are above 98%, again confirming that one may be comparing oranges with apples. Simply using a random effects model and providing prediction intervals doesn't overcome this problem. The sentence 'indicating that a new potential observation in a similar setting would likely yield a proportion of patients receiving antibiotics close to 50%' is therefore also problematic.

22. All antibiotic prescriptions were classified as inappropriate in three studies. This is for the Chinese study likely based on inappropriate methodology to determine inappropriateness. E.g. RTI with a likely viral etiology, can still include a certain proportion of patients that would legitimately receives antibiotics, e.g. see https://academic.oup.com/jac/article/73/suppl_2/19/4841820 & https://academic.oup.com/jac/article/73/suppl_2/ii11/4841821

23. 'however Watch-group antibiotics accounted for high proportions among Mexico (90.3%). This extremely high percentage doesn't really line up with other studies (on restrictive populations): https://www.ncbi.nlm.nih.gov/pubmed/30522834. Of course this is a different population, but I find it at least surprising if use of Watch antibiotics would be more common among less severely ill patients in Mexico. Furthermore, the 90% is also way off the % estimated in another recent study: https://www.thelancet.com/journals/laninf/article/PIIS1473-3099(18)30547-4/fulltext

24. 'The WHO recommends that the proportion of patients receiving antibiotics in an outpatient setting should be less than 30% [68]. However, this threshold was estabilished somewhat arbitrarily more than two decades ago, due to the lack of evidence on prescription practices and actual needs according to patient's clinical features.' There are examples in UK and EU where condition-specific ideal prescribing proportions were estimated (and compared with real prescribing data): 

https://academic.oup.com/jac/article/73/suppl_2/19/4841820 & https://academic.oup.com/jac/article/73/suppl_2/ii11/4841821

https://qualitysafety.bmj.com/content/20/9/764

25. 'A recent published umbrella review'. Please specify that this focused on dental care.

26. 'or include thorough laboratory testing to match diagnosis with antibiotic use'. This would likely result in unfair comparisons as such testing is typically not available (not even in HIC setting) and in face of uncertainty it would be acceptable that a certain proportion of patients would be treated with antibiotics even if it would not be correct according to strict laboratory tests. 

Reviewer #2: Interesting review, I have a few comments regarding the selection of the studies:

1)The authors said "Conference proceedings and abstracts, commentaries, editorials, reviews, mathematical modelling studies, economic analyses, qualitative studies, and studies published in predatory journals as defined by Beall [13] were excluded": Why is the rationale to exclude Conference proceedings and abstracts? sometimes these studies never get a full manuscript, because the researchers didn't have enough time, so they publish in conference abstract book. Of course, these studies have higher risk of bias and low quality, but can provide extra-evidence. Similar situation are papers from predatory journals (n=5): why not evaluate them and report the quality of them? Sometimes some researchers (specially at LMICs) didn't know the problem regarding predatory journals.

2)Figure 2: I'm a bit surprise about the low risk of "Reliability & validity of method used for measuring prescriptions" and "avoidance of inappropriate exclusions", because one of the most frequent data source were medical records and, in general, the quality from these records is not too high. I suggest to discuss a bit about it.

3)Figure 3: I suggest to don´t present an overall OR, at least you can present in sub-groups based in the type of denominator. I suggest only present as was presented in the supplementary material, because the study population is different

4)Meta-regression and heterogeneity: I suggest to evaluate the studies used for meta-regression, maybe only use studies with similar denominator

Reviewer #3: This is an important study that has the potential to contribute in informing policy and practice in tackling antimicrobial resistance as it attempted to analyse prescription practice in LMICs. It is properly conducted and very well written. The dearth and poor quality of the studies in these settings as demonstrated by the review limits the drawing of any strong conclusion as stated by the authors on the practice and appropriateness of antibiotic prescriptions. More importantly, the interpretation of such results should be carefully crafted and contextualized as every year inadequate access to antibiotics kills nearly 6 million people, including a million children who die of preventable sepsis and pneumonia mostly in LMICs. Therefore, interpretation of the data requires caution in the face of this reality

Major comments:

1. The operational definition of primary care as given by the authors is the foundation of the review. However, the definition is bundled into non-specific categorisation that its application limits ultimate utility of the study for any meaningful policy and practice implications. The level of tier of the health system (e.g. referral/tertiary/provincial/district hospital; health center/polyclinic/health station/clinic) in a country determine and regulate the system that determine prescription practice (e.g. the cadre of health worker; type and class of medicines/antibiotics should be available and used). Therefore, presenting the data by specific health facility would help to inform any potential antimicrobial/antibiotic stewardship programmes in LMICs and enhance the relevance and utility of the study. 

2. The WHO AWaRe classification of antibiotics as the authors mentioned is to assist antibiotic stewardship efforts and promote optimal use so as to prevent the development of drug resistance. The classification underline that the narrow-spectrum antibiotics in the Access group are the preferred treatment option for most infections (e.g Respiratory tract) and are also thought to have a lower ecologic impact regarding the selection and spread of antibiotic resistance than broader-spectrum agents. Therefore, Access group antibiotics should therefore constitute the majority of antibiotic use in the outpatient setting and overall. As part of the outcome measurement of the current Global Programme of Work of WHO, countries should strive to ensure that Access group antibiotics constitute more than 60% of the overall antibiotic use. Authors should frame their analysis, presentation and interpretation likewise. 

3. While recognising the paper will undergo statistical review, I wonder on the value of the pooled estimates. 

Minor comments: 

* Authors' use of 'community' level in the paper has to be clarified or removed. 

* Clarify and correct use of antimicrobial vs antibiotic in the text

* Ref 68 is obsolete and need to be removed 

Reviewer #4: I confine my remarks to statistical aspects of this paper. 

These were well done and I recommend publication.

Peter Flom

[LINK]

---

## [Decision Letter · Decision Letter 2]

22 Apr 2020

Dear Dr. Pai,

Thank you very much for re-submitting your manuscript "Antibiotic prescription practices in primary care in low- and middle-income countries: a systematic review and meta-analysis" (PMEDICINE-D-20-00248R2) for review by PLOS Medicine.

I have discussed the paper with my colleagues and the academic editor and it was also seen again by three reviewers. I am pleased to say that provided the remaining editorial and production issues are dealt with we are planning to accept the paper for publication in the journal.

[LINK]

We look forward to receiving the revised manuscript by Apr 29 2020 11:59PM. 

Sincerely,

Adya Misra, PhD

Senior Editor 

PLOS Medicine

plosmedicine.org

Requests from Editors:

Abstract

My apologies for the ambiguous request regarding abstract structure. We ask that you include the relevant information provided here within the PLOS Medicine abstract subheadings of background, methods and findings, conclusions. Therefore, the background section should contain the aim of your study whereas the remainder of the information should be provided within methods and findings, followed by conclusions. The last sentence of the methods and findings section should include the limitations of your study. 

Author summary

Lines 101-102 require some clarification since the access group is not a widely used term. Could you please simplify or introduce the AWaRe classification in the author summary?

Line 347 could you use an alternative term to “all-comers”

Please update the bibliography in Vancouver style 

PRISMA checklist- please use sections and paragraphs instead of page numbers as these are likely to change during publication

Comments from Reviewers:

Reviewer #1: Virtually all comments are addressed.

However, the sentence 'Only nine studies assessed rationality, and the proportion of inappropriate prescription ranged from 8% to 100%.' from the abstract is still a bit misleading. 

Could the authors please add that this is for various specific conditions? There is unlikely a country in the world where antibiotics are only prescribed inappropriately, for specific symptoms/conditions yet, but not for all conditions/symptoms.

Reviewer #3: The points I raised are adequately addressed or explained. I have no further comment

Reviewer #4: I confine my remarks to statistical aspects of this paper.

I had already approved it, so I recommend publication

Peter

[LINK]

---

## [Editor Report · Decision Letter 3]

8 May 2020

Dear Dr. Pai, 

On behalf of my colleagues and the academic editor, Dr. Margaret Kruk, I am delighted to inform you that your manuscript entitled "Antibiotic prescription practices in primary care in low- and middle-income countries: a systematic review and meta-analysis" (PMEDICINE-D-20-00248R3) has been accepted for publication in PLOS Medicine. 

PRODUCTION PROCESS

PRESS

PROFILE INFORMATION

Thank you again for submitting the manuscript to PLOS Medicine. We look forward to publishing it. 

Best wishes, 

Adya Misra, PhD

Senior Editor 

PLOS Medicine

plosmedicine.org